# Rapid saturation of cloud water adjustments to shipping emissions

Peter Manshausen[1], Duncan Watson-Parris[1,2], Matthew W. Christensen[3], Jukka-Pekka Jalkanen[4], and Philip Stier[1]

[1]Atmospheric, Oceanic and Planetary Physics, Department of Physics, University of Oxford, Oxford OX1 3PU, UK
[2]now at: Scripps Institution of Oceanography and Halicioğlu Data Science Institute, University of California San Diego, La Jolla, CA, USA
[3]Pacific Northwest National Laboratory, Richland, WA, USA
[4]Finnish Meteorological Institute, Helsinki, Finland

**Correspondence:** Peter Manshausen (peter.manshausen@physics.ox.ac.uk)

**Abstract.** Human aerosol emissions change cloud properties by providing additional cloud condensation nuclei. This increases cloud droplet numbers, which in turn affects other cloud properties like liquid water content, and ultimately cloud albedo. These adjustments are poorly constrained, making aerosol effects the most uncertain part of anthropogenic climate forcing. Here we show that cloud droplet number and water content react differently to changing emission amounts in shipping exhausts. We use information about ship positions and modelled emission amounts together with reanalysis winds and satellite retrievals of cloud properties. The analysis reveals that cloud droplet numbers respond linearly to emission amount over a large range (1-10 kg h$^{-1}$), before the response saturates. Liquid water increases in raining clouds, and the anomalies are constant over the emission ranges observed. There is evidence that this independence of emissions is due to compensating effects under drier and more humid conditions, consistent with suppression of rain by enhanced aerosol. This has implications for our understanding of cloud processes and may improve the way clouds are represented in climate models, in particular by changing parameterizations of liquid water responses to aerosol.

## 1 Introduction

The effect of aerosols on cloud radiative properties contributes the largest uncertainty to estimates of anthropogenic climate forcing (Masson-Delmotte et al., 2021). A large part of this uncertainty is from the adjustment of liquid water content, quantified by column liquid water path (LWP), to increased numbers of cloud droplets ($N_d$)(Gryspeerdt et al., 2019a). One line of evidence used to constrain this relationship are so-called opportunistic experiments (Toll et al., 2019), where a pollution source allows a direct comparison of otherwise similar clouds under polluted and unpolluted conditions. A recent review of these is given by Christensen et al. (2021). Among the most striking opportunistic experiments are ship tracks (Conover, 1966; Durkee et al., 2000; Schreier et al., 2007; Wang et al., 2011), long, linear cloud features where ship emission aerosols have increased cloud albedo so that they can be identified in satellite imagery.

While ship track studies allow to study aerosol-cloud interactions in isolation, they introduce biases: With respect to space, Possner et al. (2018) suggest that ship tracks are mostly sampled from shallow boundary layers (<800 m), because they are not

often visible in deeper ones. Using large eddy simulations (LES) of deep boundary layers they find tracks that are hidden in natural variability, by averaging along track. Simulations (Possner et al., 2020) and satellite observations (Chen et al., 2012) show a stronger LWP decrease in deeper boundary layers. With respect to time and still using LES, Glassmeier et al. (2021) find that entrainment reductions in LWP occur on timescales of ca. 20h. They argue that by this time, most ship tracks are broken up and that ship track studies therefore underestimate the negative LWP response in the "observed visible" ship tracks occurring in non-precipitating clouds.

Addressing the spatial selection bias, Gryspeerdt et al. (2019b) and Diamond et al. (2020) compare entire regions of high and low shipping emissions, while Watson-Parris et al. (2022) compare higher and lower emission years because of regulatory changes. Gryspeerdt et al. (2021) use ship positions and emissions data to show a time-resolved picture of aerosol-cloud interactions in ship tracks. In previous work, we demonstrated a strongly positive LWP response in "invisible" ship-tracks in the trade cumulus (outside of the classic Stratocumulus deck regions), relying only on advected emissions and not on logging visible tracks in satellite images (Manshausen et al., 2022). This provides evidence that the selection biases discussed above have an important effect on the observed aerosol-cloud interactions.

## 2 Emission influence on cloud properties

As in Manshausen et al. (2022), we study shipping effects using data of ship positions. We simulate where their emissions are advected to by the time of satellite overpass, collecting MODIS data, and compare the in-track and out-of track cloud properties in these locations (see Methods for more details). Our study region is bounded by (-50°S, 50°N) and (-90°W, 20°E), see Fig 3 for a map view. Compared to traditional ship track studies, this method does not rely on hand-logging clouds with decreased droplet radii, commonly identified from the infrared channels like the 3.7 $\mu$m band of MODIS (Coakley and Walsh, 2002; Segrin et al., 2007; Christensen and Stephens, 2011). Therefore, it also does not introduce a sampling bias for the conditions that allow aerosol emissions to reduce droplet radii in a clearly discernible, linear cloud feature.

Here, we combine the data of cloud property changes in polluted locations with data on the amount of $SO_x$ emitted in these locations. This data is from the Ship Traffic Emission Assessment Model (STEAM) of the Finnish Meteorological Institute (see Methods) (Jalkanen et al., 2012; Johansson et al., 2017).

Combining the emission and satellite data, we can establish the relationship between emissions and cloud property changes. Figure 1 shows the in-track enhancements of $N_d$ and LWP over the time since emission. The signal shows a maximum around two to four hours after emission, then a slow decline falling to zero around 15 hours. A number of processes are likely responsible for the decline back to zero: The mixing of the track with surrounding air, the onset of precipitation, and the more uncertain location of the retrieved position with longer advection times. The emission amount controls the response in $N_d$, with the enhancements being larger the larger the emissions are. The same is not true for the LWP response, which seems insensitive

to the emissions amount. For all emissions quartiles, the response increases from zero and reaches a stable state after around five hours. The level to which it increases is the same for the all quartiles.

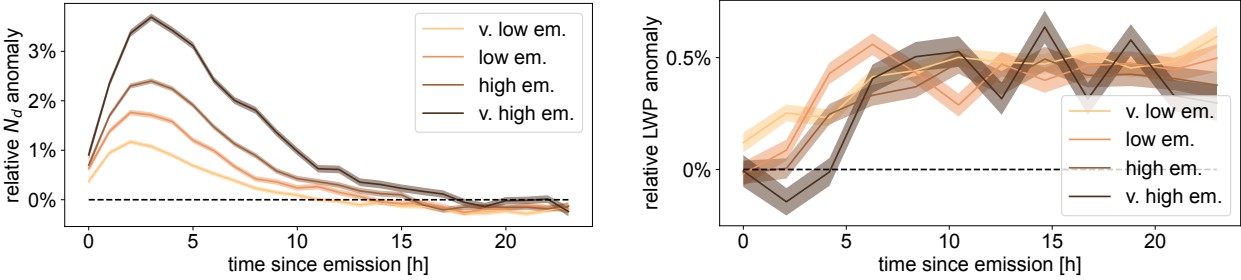

**Figure 1.** Cloud property responses to ship emissions over time, by emission quartiles. The four equally populated emission bins are labelled very low to very high. This data is from averaging all the data in-track that falls into a time and emission bin, and dividing by the average of all the out-of-track data in the same time and emissions bin.

## 3 IMO regulation change

In 2020, the International Maritime Organization (IMO) introduced strict limits on shipping fuel sulphur content, limiting it from 3.5 to 0.5% by mass. In some regions, such as off the North American coast and in the North Sea, more stringent limits of 0.1% have been in place since 2015. However, these areas are small compared to our study region, so the 2020 emissions change presents a valuable opportunity to study the effect of low shipping emissions on clouds. Furthermore, compared to other studies on emissions regulation, such as those by Gryspeerdt et al. (2019b) or Watson-Parris et al. (2022), we do not rely on the emissions causing a ship track visible to the eye. Recently, Diamond (2023) has shown the impacts of these changes independently of ship tracks in a shipping corridor in the Southeast Atlantic, with a complimentary approach to this study.

Figure 2 shows the change in $SO_x$ emissions between 2014 (pre-IMO regulation change) and 2021. Fuel sulphur content was reduced by 80% after the regulation and this is reflected in $N_d$: While $N_d$ enhancement reaches more than 2% before, it stays well below 1% after 2020. However, LWP adjustments are of the same magnitude in both cases. This is very similar to the responses by emission quartiles shown in Figure 1.

Figure 3 shows the pre- and post 2020 responses over space in the study region. 3c) and d) are 2014 to 2019 as in Manshausen et al. (2022), whereas a) and b) are for 2021, and e) and f) show difference plots. The Stratocumulus regions of the South East Pacific and Atlantic above the cold waters of Eastern boundary upwelling regions are those with the strongest $N_d$ anomalies in a) and c). The most positive LWP response is in the Atlantic trade Cumulus. As with the time-resolved signal in Figure 2, the response in $N_d$ is different before and after 2020, with the difference plot in e) showing decreases in $N_d$ anomaly

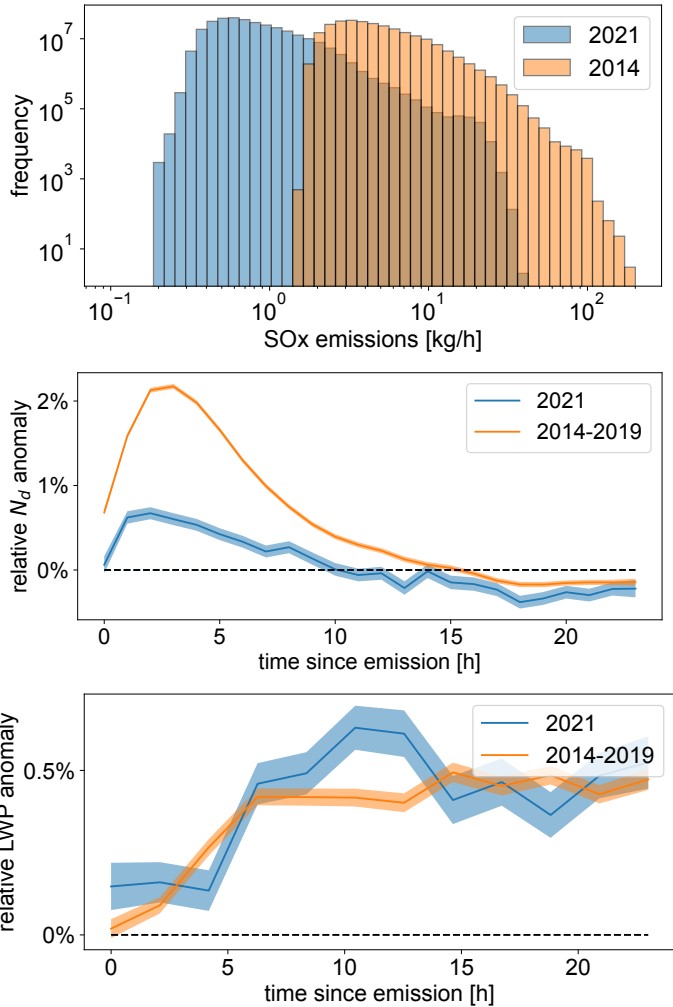

**Figure 2.** Cloud property responses to IMO fuel sulphur regulation in 2020. Shown is a histogram of sulphur emissions of ships in the study region from the STEAM model, comparing 2014 and 2021. b) and c) show the responses of cloud properties $N_d$ and LWP, each time comparing the six years before with the year after 2020, when the fuel sulphur content decrease was mandated by the IMO.

almost everywhere. These are most pronounced where the $N_d$ anomaly was large before. The LWP response, again, does not seem to change systematically after 2020, with the difference plot in f) showing no spatially coherent changes.

## 4 Dependence on drizzle conditions

If the enhancement in LWP is due to drizzle suppression, then it can be expected to depend on the background droplet radius, as shown by Toll et al. (2017). They use a threshold of droplet effective radius of $15\mu$m. Above $15\mu$m we expect the presence

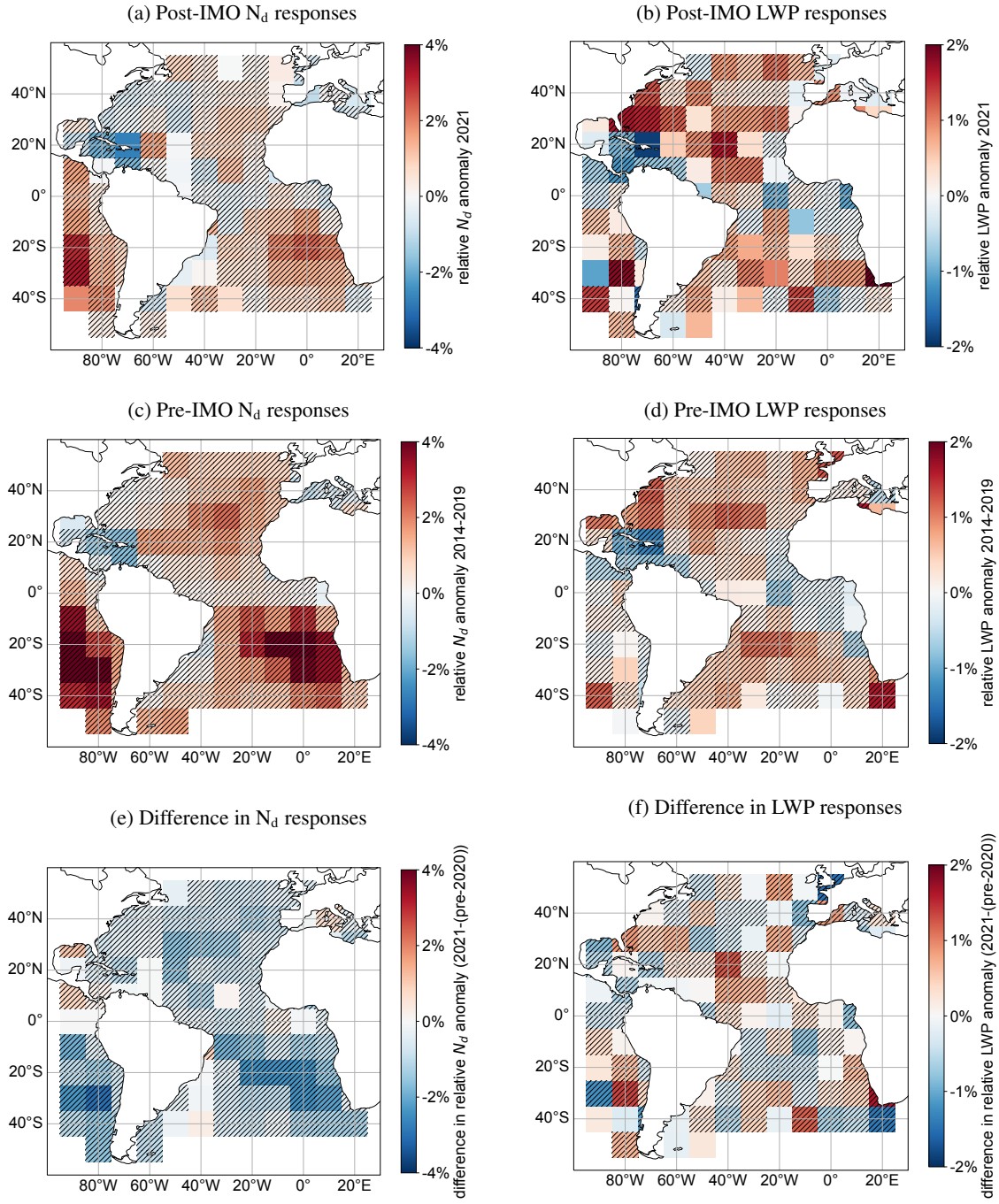

**Figure 3.** Comparison of regional patterns of $N_d$ and LWP responses before and after sulphur emissions regulation. Heatmaps show the responses in $N_d$ and LWP averaged over 10h after emissions for $N_d$ and 24h for LWP. The top row shows the post-regulation responses, the middle one the pre-regulation responses, and the bottom row the differences. Note that the colorbar ranges are different for $N_d$ and LWP. Hatching in a)-d) incicates statistically significant differences ($p < 0.1$ in a two tailed Student's t-test) from the null experiment (see Methods). In e) and f) we test for significant differences between the pre- and post IMO anomalies. The bottom row has too little ship traffic to collect data.

of drizzle, increased gravitational settling and subsequent LWP loss through precipitation. This is the regime where additional aerosol can decrease droplet radii to suppress precipitation and maintain high LWP. Sorting the data by emission quartiles and then into droplet radii below or above $15\mu$m, we obtain the LWP evolutions of Figure 4. As hypothesized, they do not show clear enhancements in the no drizzle case. In the drizzling case however, LWP is enhanced in the track. This supports the mechanism of drizzle suppression for LWP increases.

We can resolve the emission-dependence of $N_d$ and LWP, when we do not also stratify by time since emission. Figure 5 shows the emission dependence of $N_d$ and LWP. For $N_d$, we give the enhancement averaged over the first ten hours after emission, as this gives a stronger signal (compare time-resolved plots like Figure 1). In the bottom plot we show the number of data points corresponding to a bin of $SO_x$ emissions. The small, blue low emission peak represents the 2021 data and the larger, orange one the data between 2014 and 2019. Note, that in the top panel the orange line includes both 2021 and 2014-2019 data. The response of $N_d$ to emissions is roughly linear over a large range of emissions, well above and below the pre-2020 mean. The response saturates for very high emissions past ten kg/h, and it may show nonlinear behaviour for very low emissions, cases which are somewhat more uncertain due to the lower number of data points.

Figure 5 also shows the emission dependence of LWP enhancements, for all data and for the subsets where LWP is above or below 100 gm$^{-2}$. Results by Suzuki et al. (2015) suggest 100 gm$^{-2}$ of LWP separates higher LWP drizzling and raining from lower LWP non-precipitating clouds. In a similar way as for the stratification by effective radii in Figure 4, the figure shows LWP increases in the precipitating case. These increases depend on emission, but far less than those of $N_d$. Similarly, in the non-precipitating case, there are LWP reductions that become more important with increasing emissions.

The all-data curve is almost independent of emissions, consistent with the results from the previous sections on quartiles and IMO regulation. For low-emission cases 'all data' is even above the high-LWP case. This can be explained because counter-intuitively, 'all data' is not exactly the combination of the other two subsets. To avoid regression to the mean biases, we require both the in- and out-of-track to fulfill the same conditions (see Methods). This means that 'all data' includes those cases where the out-of-track LWP is lower and the in-track LWP higher than 100 gm$^{-2}$, not present in either subset. The results show that the LWP response is pronounced even in the lowest observed emissions bins.

## 5 Free tropospheric relative humidity

Humidity above cloud is correlated with the cloud property anomalies. Background rainy conditions are necessary for LWP enhancements through precipitation suppression, while LWP reductions are due to cloud top drying. This hinges on the entrainment of dry air, and we therefore expect the relative humidity in the free troposphere to play a role in the overall cloud response to aerosol. For instance, Glassmeier et al. (2021) (supplementary material) showed negative LWP adjustments in large eddy simulations at free-tropospheric water vapor mixing ratio of less than 2.8 g/kg. Figure 6 shows the enhancements as a

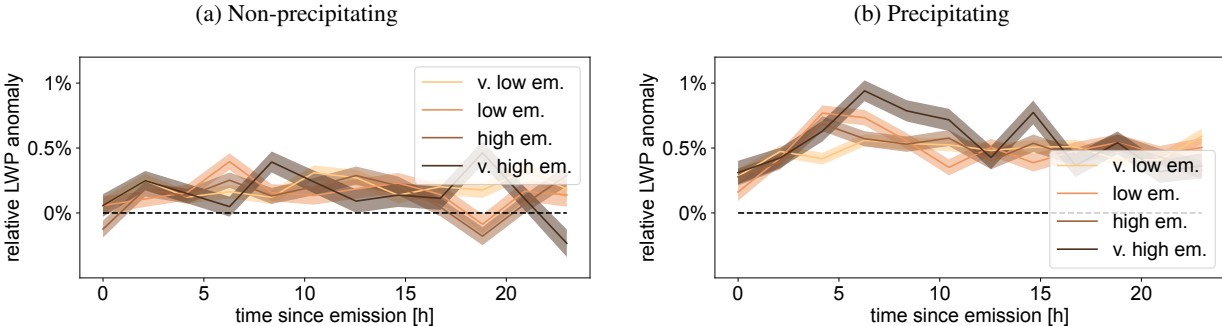

**Figure 4.** LWP only strongly increases in rainy conditions. LWP responses to ship emissions over time, by emission quartiles. The left plot shows non-rainy conditions, with droplet radii smaller than $15\mu$m, the right plot shows rainy conditions, radii greater than $15\mu$m.

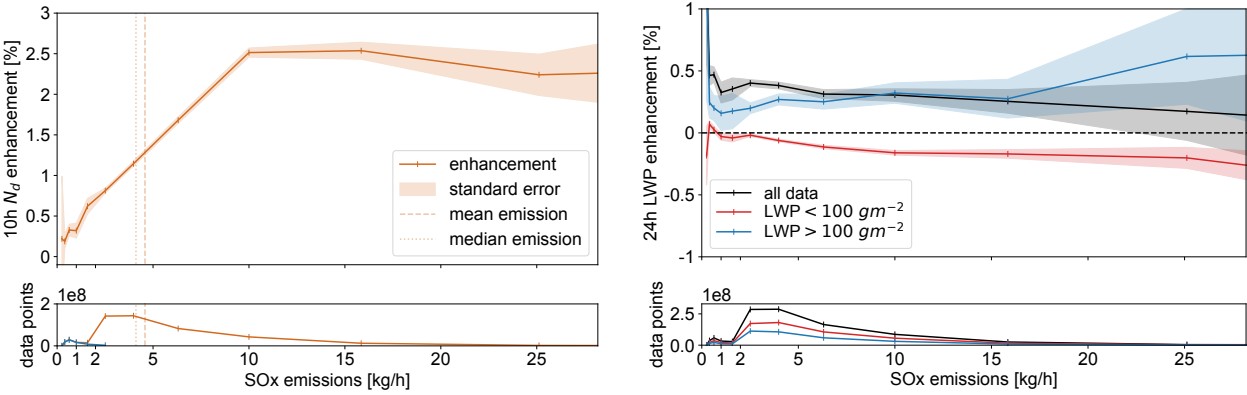

**Figure 5.** Cloud property responses to ship emissions emission amount. Shown on the left are the $N_d$ response over the first ten hours since emission as a function of emission amount. Inset in the bottom is the number of data points at the given emission amount. Shown on the right in a similar way are the LWP response, for all data, as well as stratified by LWP conditions, and again the number of points in the bottom.

function of relative humidity at 700 mbar RH700 (a proxy for the – pressure variable – above cloud relative humidity. There is a clear covariation of the two, as shown in a) and b), where we stratify the $N_d$ and LWP enhancements over time into four RH700 quantiles. The drier the air above cloud, the stronger the droplet number enhancement. The LWP enhancements behave the opposite way, and grow strongest for the most humid quartile. Looking at the enhancements over RH700 bins in c) and d) we see a large regional difference in $N_d$, with the Stratocumulus showing the largest increase. A difference in cloud regimes may help to explain the change in in $N_d$ responses as a function of RH700. In LWP, the Stratocumulus behaves similarly to the Azores region, with LWP enhancements small or negative at low humidities, and large at high humidities. While this is consistent with precipitation suppression and evaporation enhancement mechanisms, it is unclear what part of the covariability is causal. For instance, RH700 may be correlated to other cloud controlling factors such as airmass history, aerosol loading, or updraft speeds which will also influence adjustments to aerosol perturbations.

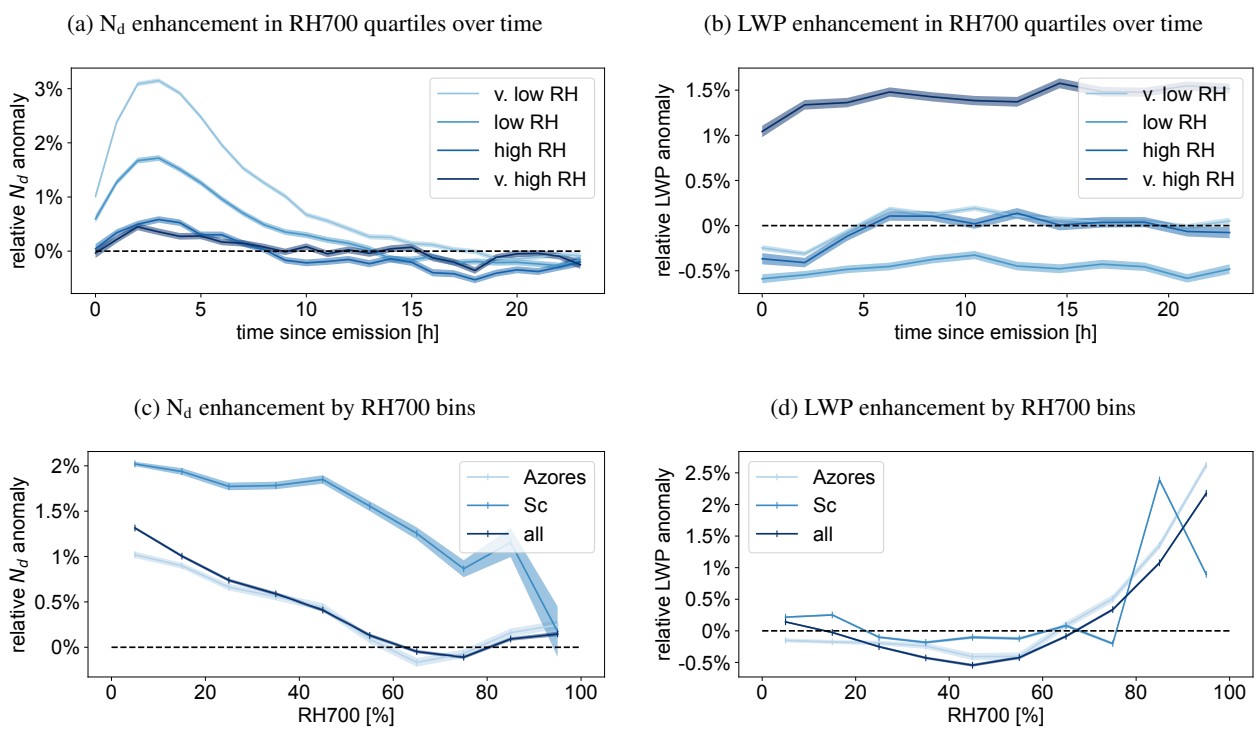

**Figure 6.** The effect of free tropospheric relative humidity (at 700mbar, RH700) on $N_d$ and LWP anomalies. The top row shows the hourly anomalies by quartiles, the bottom row averages over time and instead stratifies by region (around the Azores, and in the Namibian and Chilean Stratocumulus, Sc).

## 6   Discussion & Conclusion

We have shown the dependence of cloud properties on aerosol concentrations from shipping pollution using information of ship positions and modelled emissions together with reanalysis winds and satellite retrievals of cloud properties. Specifically, we found a strong dependence of $N_d$ enhancements on emission amounts, with more emissions leading to larger $N_d$ anomalies in tracks. Meanwhile, LWP increases of about the same magnitude are found to occur across all different emission levels.

This is also well supported by the changes from the pre- to post-2020 data, corresponding to an 80% reduction in sulphur emissions. The corresponding change in $N_d$ anomalies in tracks, much lower in 2021, confirms the changes observed between quartiles of emissions from the STEAM2 model. Similarly, LWP anomalies are largely unchanged before and after 2020, as well as between quartiles of STEAM2 emissions, lending credibility to the model. Having only one year of data for the post-IMO regulation period means that there is not as much data as for the pre-IMO regulation data. However, even a single year of ship track data produces $O(10^5)$ tracks, and results are significant with respect to standard errors. Using a single year can potentially signify different climatological conditions, with different cloud properties in 2021. However, as we are only presenting departures from background states, results would only be different if susceptibility to aerosol changed. This effect is likely small, but hard to rule out or quantify without clear knowledge of the cloud controlling factors, which we are currently investigating. There will also be a change in signal stemming from the cleaner background conditions in 2021, where the aerosol burden in shipping corridors is reduced as a cause of the lower $SO_x$ emissions of the individual ships. We find an average out-of-track $N_d$ of 65.9 cm$^{-3}$ pre-2020 compared to 64.8 cm$^{-3}$ post-2020, a difference that is significant with respect to the standard error of the mean, but likely too small to change the 'average cloud' sensitivity to aerosol perturbations. This may be different in shipping corridors, which will have experienced a larger decrease because of the high concentration of shipping.

Furthermore, we showed that the LWP enhancements preferentially occur in the drizzling regime where effective radii of cloud droplets are larger than $15 \mu m$, and LWP values above 100 gm$^{-2}$, consistent with drizzle suppression. In the non-drizzling regime we find no decreases in LWP, expected due to enhanced evaporation at cloud top, contrary to several studies of visible tracks (Segrin et al., 2007; Christensen and Stephens, 2011; Chen et al., 2012). However, no decreases were observed in our previous study including 'invisible tracks' (Manshausen et al., 2022), nor in the ship tracks shown by Gryspeerdt et al. (2019a), their Fig. 6. We find that above cloud relative humidity co-varies with LWP adjustments (Fig. 6), and that there are negative adjustments in drier conditions. The emissions dependence of increases and decreases of LWP in different regimes may compensate, offering a possible lead to explain the weak overall dependence of LWP anomalies on emission amounts. This dependence on environmental conditions is consistent with the findings of Toll et al. (2017). While Gryspeerdt et al. (2019b) find a similar independence of emissions for LWP enhancements, they find LWP enhancements only in low LWP environments. However, their data is only representative for visible tracks in the stratocumulus cloud regime while our methodology includes all polluted clouds and all cloud regimes that occur in the study region, in particular trade cumulus.

The weak dependence of LWP enhancements on emission amount is more puzzling, because we expect the LWP enhancement to be caused by the previous $N_d$ enhancement, which itself is much more dependent on emissions. A possible explanation for this is a non-linear or threshold behaviour, where even the small enhancement in $N_d$ at low emissions is enough to shut down precipitation. It is possible that we do not have enough very low emission data to observe the case where the $N_d$ response is too small to suppress precipitation. If this threshold behaviour is confirmed, it has important implications for calculations of aerosol forcing, which are currently performed using power law relations between aerosol amount, Nd, and LWP (Bellouin et al., 2020).

This can be compared to the different ways autoconversion, the process that converts cloud droplets to drizzle, is parameterized in global climate models. The threshold behavior of the initiation of autoconversion on cloud liquid water content has long been known (Kessler, 1969). In modelling, Suzuki et al. (2013) evaluate different thresholds for drizzle formation in autoconversion schemes using radar and MODIS observations. They find that the models which best match the satellite observations are the ones that use a higher threshold for drizzle formation. This means allowing drizzle formation only at effective radii larger than the threshold value. The threshold may not be just a tuning parameter, but rather represent a real, nonlinear physical process. However, simply changing autoconversion rates may not achieve the desired Nd-LWP relationship alone (Christensen et al., 2023).

We cannot exclude alternate hypotheses for the observed emissions-independence of LWP enhancements. For example, Wang et al. (2011) show that in LES of ship tracks, dynamic effects can lead to decreases in cloud cover in the out-of-track region. If this occurred in the cases we observe, we might be seeing a negative adjustment of LWP out-of-track, rather than a positive one in-track. However, our out-of-track retrievals are further away than the dark tracks observed by Wang et al. (2011), which are centered around 15 km from the middle of track compared to the distance of 30 km for our out-of-track retrievals. Furthermore, dynamic responses are also linked to aerosol amounts, and therefore should disappear at low emissions like any microphysical responses. Direct injection of water vapor without aerosol (from nuclear powered ships) showed no cloud responses in the Monterey Area Ship Track (MAST) experiment (Durkee et al., 2000) and therefore seems unlikely to play a role.

Future work will help to elucidate these questions by analyzing cloud perturbations with higher spatial resolution across-track, i.e. not only use in-track and out-of-track, but retrieve at multiple distances from the central estimate of emission locations. This should allow to discern an in-track increase from an out-track decrease of LWP. Similarly, we want to add an analysis of cloud fraction, as quantified by successful retrievals of cloud properties in the pixels across track. The pixel-level retrievals are important, as cloud fraction measures depend on the scale on which they are defined, which makes this analysis more challenging. It may be important because cloud fraction responses have been claimed to be as important as $N_d$ enhancements to the radiative effect of aerosol emissions (Rosenfeld et al., 2019).

*Code and data availability.* ERA5 data is freely available from https://cds.climate.copernicus.eu/. MODIS data is freely available from https://ladsweb.modaps.eosdis.nasa.gov/search/. Emission datasets were obtained from Jukka-Pekka Jalkanen (Jukka-Pekka.Jalkanen@fmi.fi).

The complete collocated data for ship tracks studied here including cloud property measures like $N_d$ and LWP has been archived by CEDA under DOI: 10.5285/2d0f8bb3927b4f75ae75276705858f68. Code for the production of the collocated data has been archived under DOI: 10.5281/zenodo.6556425.

## Appendix A: Methods

### A1   Scope

Geographically, we study the larger part of the Atlantic between (-50°S, 50°N) and (-90°W, 20°E) as well as the stratocumulus (Sc) deck in the Southeast Pacific off the Chilean coast. The size of the region is limited by computational cost and we choose to place it so it covers two Sc regions. Data is for the years 2014-2019 as well as 2021 (chosen to compare between the pre- and post IMO regulation case).

### A1   Retrieval at polluted cloud locations

We find ship polluted clouds in satellite data without relying on a change in cloud droplet effective radii or for the tracks to be visible to the eye. For this, we rely on ship positions from the AIS system. As the AIS data itself is proprietary, we reconstruct ship tracks from hourly emission grids provided by the Finnish Meteorological Institute. These are heatmaps at 0.005°spatial resolution. Ship position points are found using the trackpy library Allan et al. (2019). These positions, interpolated to 5min intervals, are then used as input to the HYSPLIT model Stein et al. (2015). The model uses ERA-5 reanalysis data (which

was converted into ARL format) together with the initial position and an assumed emission height of 20 m to simulate the location of the advected emissions up to 24 h after emission (this is on the lower end of the emission height of most ships, but produces the best-matched tracks in visual inspection of test days with visible tracks). Out-of-track retrievals are taken by using the shape of the advected emissions track, but 30 km to either side of the track. This is done by calculating the 'direction' of a track, taken as the vector from start to end, then calculating the orthogonal vector, and then displacing the track

by 30 km in the orthogonal vector's direction. For a given overpass of the Aqua or Terra satellites, which carry the Moderate Resolution Imaging Spectroradiometer (MODIS) Platnick et al. (2003), the positions of the emissions at this time is selected and collocated to the MODIS Level-2 cloud product MOD/MYD 06, collection 6.1. $N_d$ is obtained following Quaas et al. (2006) from retrievals of cloud optical thickness and effective radius (see also the derivation to eq. 11 in Grosvenor et al. (2018)). Compared to hand-logged tracks, the retrieval locations are less certain, owing to uncertainty in the reanalysis wind

data used to advect the emissions. This means that overall the enhancements are smaller compared to natural variability, i.e. a lower signal-to-noise than in ship track studies. This problem is tackled with large amounts of data, as well as careful sampling to avoid introducing any biases (see below for conditioning on retrieved properties).

## A2 Input for modelling ship emissions

The Ship Traffic Emission Assessment Model of the Finnish Meteorological Institute (STEAM) uses as input the following datasets: a) Vessel activity. This consists of global Automatic Identification System (AIS) transponder messages, which include data both from terrestrial and satellite AIS networks. Global vessel activity datasets are provided by commercial operators and restricted to FMI research purposes. These are currently provided by Orbcomm Ltd. b) Global fleet description. These data include technical features of all the ships in the global fleet and are provided by a commercial operator. Required data include physical dimensions, machinery, propulsion system details, power generation and transmission features, capacity description and installation of emission abatement techniques (e.g. exhaust gas cleaning systems). In this case, data from IHS Markit and IMO GISIS are used. c) Polygon descriptions of special areas. These include, for example, Emission Control Areas (ECAs) for air emissions. These input datasets define, for each vessel, its capabilities of using various fuels during the modeling runs. These are defined by engine properties, operation area and time stamp (for entry into force of regulation). Environmental conditions such as sea current and surface wind speed and direction can in principle be added to the model. Here, they are not considered because they would slow the computation so that the resolution would need to be lowered.

## A3 Emissions model

We are giving a high-level description of the STEAM calculation process. For the details of the calculation process, see Jalkanen et al. (2012); Johansson et al. (2017). In STEAM, vessel resistance is determined by the following components.

$$R_{total} = R_{residual} + R_{friction} + R_{fouling} \tag{A1}$$

Once R$_{total}$ is known, the necessary engine power $P$ is determined from

$$P = \frac{R_{total} v_{inst}}{q} \tag{A2}$$

in which the $q$ is the quasi-propulsive constant which includes the propulsive losses of power transmission and propeller efficiency. The calculation is described by Watson (2002) and includes contributions from propeller rotation speed and vessel length. Additional components to performance prediction include the effect of waves and sea current, but these are considered as modifications of vessel speed (knots), not resistance (kilonewtons). The individual resistance components of R$_{tot}$ are modeled using the following methods: R$_{residual}$ comes from the Hollenbach resistance prediction Jalkanen et al. (2012), which is a parameterized model based on resistance tests of 433 vessels and considers e.g vessel hull shape, bulbous bows, and different resistance components. R$_{friction}$ and R$_{fouling}$ uses the ITTC method to determine hull friction and fouling, respectively ITTC (2017) .

## A4 Conditioning on background effective radii and LWP

The data of advected emissions tracks is noisy. Therefore, subsetting the data set following any kind of criterion needs to be done very carefully. Consider the case where we would like to look at precipitating conditions as indicated by effective radii.

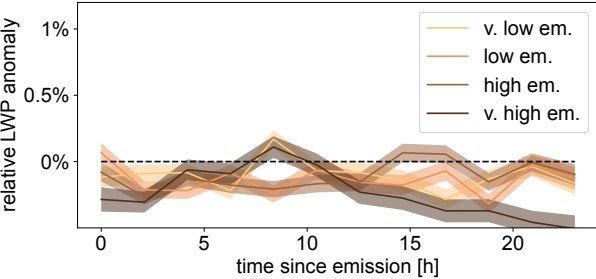 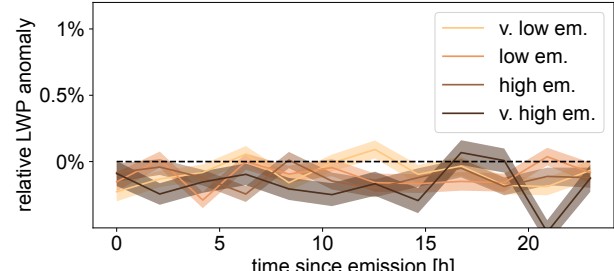

**Figure A1.** No significant LWP response in a control experiment. Here, retrievals are done with the same geometry as before, but using the positions of ships that have not yet passed through the retrieval region. The left plot shows non-rainy conditions, with droplet radii smaller than $15\mu m$, the right plot shows rainy conditions, radii greater than $15\mu m$.

Naively, we would subset the data such that in our subset the out-of-track regions have $r_{eff} > 15\mu m$. However, given noisy observations (random errors from imperfect collocation) this would introduce a bias. If there were no effect of the ship, but the data was following the same distribution in the track and the out-of-track, 'cutting off' the lower effective radius part of the out-of-track distribution would lead to different means of the track and the out-of-track. Instead, we have no choice but to require both the out-of-track *and* the track to have $r_{eff} > 15\mu m$. Stratifying this way for the below and above threshold cases, however, excludes all those cases where the track and background are on different sides of the threshold—arguably the situation we would like to observe, where precipitation that is going on outside of the track is shut down within. This is mitigated by the large size of the retrieval area (20 km across), where ship tracks should be smaller than this. Then, even if in the track effective radii are reduced, the mean of the retrieval may still be above the threshold in a precipitating regime.

## A5 Null experiments

To validate our method and to assure that increases in $N_d$ and LWP are indeed linked to the shipping, rather than the result of a bias, we also perform a null experiment. This consists in retrieving at the locations of the advected tracks, but in the satellite data of the day before the ships pass through. In the collocation of advected emissions with satellite data described above, we select the emissions that were emitted up to 24h before each satellite overpass, find their positions at the overpass time, and collocate. If we change this to use the emissions that occur in the 24h after the overpass, and advect them until the overpass time on the next day, this gives us a a sampling strategy that will show the same kind of retrieval geometry, but without the pollution from individual ships. Here, we expect no signal, and indeed, in Figure A1, treating the data otherwise in the same way as in the experiment, i.e. stratifying by emission quartiles and effective radii, we see no strong response, possibly even a small decrease. This may be due to natural variability, or the effects of other ships preferentially taking the same routes (shipping corridors), and this signal should also be present in the actual observations.

*Author contributions.* P.M., P.S., and D.W.P. developed the concept of the study and designed its implementation. J.-P.J conducted the STEAM simulations. M.C. converted meteorology files for use with HYSPLIT and provided code for interfacing with HYSPLIT. P.M. wrote
the code for collocating the data sets and analysed the data. All authors contributed to the interpretation of the results. P.M. drafted the manuscript with contributions and review from all co-authors.

*Competing interests.* None

*Acknowledgements.* This work was funded by the European Union's Horizon 2020 research and innovation programme under Marie Skłodowska-Curie grant iMIRACLI (agreement No 860100). This research was supported by the European Research Council project RE-
280 CAP under the European Union's Horizon 2020 research and innovation programme (grant no. 724602), by the FORCeS project under the European Union's Horizon 2020 research programme with grant agreement no. 821205, and by the EMERGE project with grant agreement no. 874990. D.W.P. and P.S. were supported by the UK Natural Environment Research Council project ACRUISE (NE/S005099/1). M.W.C would like to acknowledge support from the U.S. Department of Energy Office of Science Biological and Environmental Research as part of the Atmospheric Systems Research (ASR) program and the Pacific Northwest National Laboratory which is operated by Battelle for the U.S.
Department of Energy under Contract DE-AC05-76RLO1830. Analysis was performed and data stored with infrastructure provided by the UK Centre for Environmental Data Analysis CEDA. Thank you to Ed Gryspeerdt for helpful discussions. We would like to acknowledge the NOAA Air Resources Laboratory (ARL) for the provision of the HYSPLIT transport and dispersion model used in this publication. Finally, we gratefully acknowledge the valuable and constructive feedback of two anonymous referees.

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
