# Peer review of "Rapid saturation of cloud water adjustments to shipping emissions"

_EGUsphere, 2023_

## Referee Comment (RC1)

**Review of "Rapid saturation of cloud water adjustments to shipping emissions" by Manshausen et al. (egusphere-2023-813)**

The study analyzes cloud microphysical changes over the Atlantic and parts of the Pacific using satellite and reanalysis data. The authors focus on the difference between regions influenced by ships, so-called ship tracks, and the surrounding regions. Furthermore, the authors use a recent regulation that significantly reduced the amount of sulfate in ship emissions, and hence the likelihood of ship emissions producing cloud condensation nuclei. The authors find that ship emissions rapidly increase the cloud droplet concentration, which then slowly adjusts to the out-of-track value. The liquid water is only marginally affected by emissions, and only if the emissions increase the cloud droplet concentration sufficiently to inhibit precipitation. Overall, this is an interesting and relevant study. The manuscript is well-written, but it requires some clarifications. While I have one slightly major comment, I fully support the manuscript's publication in ACP Letters once my concerns are addressed.

**Major Comment**

The authors constrain liquid water path (LWP) changes by the effective radius (smaller/larger than 15 μm) and the LWP (smaller/larger than 100 g/m2). These are important steps to understanding precipitation inhibition, which results in positive LWP adjustments. However, the authors do not find negative LWP adjustments. Many authors suggest these negative adjustments are only possible if the entrained air is sufficiently dry. Thus, I recommend adding an additional constraint to the analysis: the free-tropospheric humidity. Glassmeier et al. (2021) showed negative LWP adjustments for simulations with a free-tropospheric water vapor mixing ratio of less than 2.8 g/kg.

**Minor Comments**

L. 8: To what does "this" refer to? The prior sentence writes about raining clouds, this sentence addresses "rainy and non-rainy conditions". Please clarify.

Ll. 41 – 41: Is logging "decreased droplet radii" the only way to identify ship tracks by hand?

Ll. 50 – 51: What is the reason for the slow decline in the droplet concentration anomaly? Mixing of the track with its surroundings? Precipitation scavenging? Broadening of the track?

Ll. 62 – 63: How much does the water vapor in ship emissions affect LWP adjustments? The amount of water vapor in the emissions should not depend on the IMO regulations.

Ll. 135 – 137: Kessler (1969) already suggested that autoconversion should depend on a threshold.

**Technical Comments**

Ll. 32, 39: Check citation style.

L. 21: Who is "they"? I guess you refer to ship tracks.

L. 41: I recommend mentioning the sampled regions in the manuscript's main text.

Ll. 56 – 58: How do these two sentences relate?

Ll. 66 – 67: While the caption states when the anomalies are determined, I suggest adding this to the main text.

Figs. 3 and 4: Put the title of the panels over the figure.

Fig. 5, top-left panel: Where is the blue line?

**References**

Kessler, E., 1969: On the Distribution and Continuity of Water Substance in Atmospheric Circulations. Meteor. Monogr., No. 10, Amer. Meteor. Soc., 84 pp.

---

## Author Response (AR1)

We would like to thank the two anonymous referees for their constructive and insightful comments. We are pleased you found our manuscript to be "interesting and relevant", "well-written", and "concise"; and to follow "a clear storyline". Please find detailed responses to the individual comments below.

**Referee #1**
**Major Comment**
**[...] Thus, I recommend adding an additional constraint to the analysis: the free-tropospheric humidity.**
*This is a very valuable comment. We have performed additional analysis of the data stratified by relative humidities above cloud (reanalysis RH700). This shows a strong correlation of LWP adjustments and RH700, with negative adjustments at low and in particular intermediate-low above cloud humidities. We have added the corresponding section to the manuscript.*
*Added: Section on free tropospheric relative humidity (ll. 108 ff)*

**Minor Comments**

**L. 8: To what does "this" refer to? The prior sentence writes about raining clouds, this sentence addresses "rainy and non-rainy conditions". Please clarify.**
*This refers to the independence of emissions, which come from balancing, emissions-dependent effects under high and low LWP conditions.*
*Added: independence of emissions, replaced 'rainy and non-rainy' with 'dry and more humid'*

**Ll. 41 – 41: Is logging "decreased droplet radii" the only way to identify ship tracks by hand?**
*Decreased droplet radii show very clearly in the MODIS 3.7µm retrievals, which were used in the previous studies cited below. I do not know of any other approaches that people have used in recent years (e.g. just using the visible reflectances/albedo increases). Added: "commonly identified from the IR channels like the 3.7 µm band of MODIS (Coakley and Walsh, 2002, Segrin et al. 2007, Christensen and Stephens, 2011)"*

**Ll. 50 – 51: What is the reason for the slow decline in the droplet concentration anomaly? Mixing of the track with its surroundings? Precipitation scavenging? Broadening of the track?**
*Probably all of these, as well as greater uncertainty at long advection times (time-integrated uncertainty of reanalysis winds). Added: "A number of processes are likely responsible for the decline back to zero: The mixing of surrounding air, the onset of precipitation, and the more uncertain location of the retrieved position with longer advection times."*

**Ll. 62 – 63: How much does the water vapor in ship emissions affect LWP adjustments? The amount of water vapor in the emissions should not depend on the IMO regulations.**
*We cannot rule this out entirely. However (added in l. 154):*
*"Direct injection of water vapor without aerosol (from nuclear powered ships) showed no cloud responses in the Monterey Area Ship Track (MAST) experiment (Durkee et al., 2000) and therefore seems unlikely to play a role."*

**Ll. 135 – 137: Kessler (1969) already suggested that autoconversion should depend on a threshold.**
*Thank you for pointing this out, I was not aware of this!*
*Added: "The threshold behavior of the initiation of autoconversion on cloud liquid water content has long been known (Kessler, 1969)"*

**Technical comments**
**Ll. 32, 39: Check citation style.**
*changed*

**L. 21: Who is "they"? I guess you refer to ship tracks.**
*changed word order to clarify*

**L. 41: I recommend mentioning the sampled regions in the manuscript's main text.**
*Done, l. 42*

**Ll. 56 – 58: How do these two sentences relate?**
*Added: "In some regions, such as off the North American coast and in the North Sea, more stringent limits of 0.1% have been in place since 2015. However, these areas are small compared to our study region, so the 2020 emissions change presents a valuable opportunity to study the effect of low shipping emissions on clouds. Furthermore, compared to other studies on emissions regulation, such as those by…*

**Ll. 66 – 67: While the caption states when the anomalies are determined, I suggest adding this to the main text.**
*l.69: Added years: 2014 to 2019*

**Figs. 3 and 4: Put the title of the panels over the figure. Fig. 5, top-left panel: Where is the blue line?**
*Changed.*
*We chose not to plot separate lines for the 2021 and other data in the top left panel, as most of the 2021 data is in the very data sparse low-emission bins. The orange line in the top left panel includes all data. Added, l.89: "Note, that in the top panel the orange line includes both 2021 and 2014-2019 data."*

**Referee #2**
**Detailed Comments**

**Fig. 3, Line 163, the southernmost row of this domain is partly white**
*Yes, these regions have very little ship traffic and therefore few data points. This makes for insufficient n to calculate means and standard errors, as for low n the assumption of independence is not fulfilled. We decided to exclude this area from the analysis.*
*Added: "The bottom row has too little ship traffic to collect data."*

**How are the out-of-track regions defined?**
*Added clarification on this (ll. 183 ff): "Out-of-track retrievals are taken by using the shape of the advected emissions track, but 30km to either side of the track. This is done by calculating*

the `direction' of a track, taken as the vector from start to end, then calculating the orthogonal vector, and then displacing the track by 30km in the orthogonal vector's direction."
Replaced all occurrences of 'control' where not referring to the null experiment with 'out-of-track'.

**Only one year of post-IMO regulation change data is available (2021). Please discuss in how far this may introduce an uncertainty when pre- and post-IMO conditions are compared.**
*This is a good question, we have added a discussion section on this.*
*Added (ll. 132 ff): Having only one year of data for the post-IMO regulation period means that there is not as much data as for the pre-IMO regulation data. However, even a single year of ship track data produces $O(10^5)$ tracks, and results are significant with respect to standard errors. Using a single year can potentially signify different climatological conditions, with different cloud properties in 2021. However, as we are only presenting departures from background states, results would only be different if susceptibility to aerosol changed. This effect is likely small, but hard to rule out or quantify without clear knowledge of the cloud controlling factors, which we are currently investigating. There will also be a change in signal stemming from the cleaner background conditions in 2021, where the aerosol burden in shipping corridors is reduced as a cause of the lower $SO_x$ emissions of the individual ships. We find an average out-of-track $N_d$ of 65.9 cm-3 pre-2020 compared to 64.8 cm-3 post-2020, a difference that is significant with respect to the standard error of the mean, but likely too small to change the `average cloud' sensitivity to aerosol perturbations. This may be different in shipping corridors, which will have experienced a larger decrease because of the high concentration of shipping.*

**Line 175 mentions "best-matched tracks". What does this mean and how is it tested?**
*We looked at a handful of days when tracks were visible and chose the parameter that gave the best agreement between visible tracks and the advected emissions.*
*Added (l. 185): "in visual inspection of test days with visible tracks."*

**Technical comments**
**Sometimes the position of the parentheses for references is not correct, e.g. in line 39, where it should read: "As in Manshausen et al. (2022), …"**
*Fixed*

**Line 118: add "their" before Fig. 6 to make clear that this is referring to Gryspeerdt et al.'s figure.**
*Added: "their"*

**Line 171: positons -> positions**
*Thank you. Fixed.*

---

## Referee Report (RR1)

**Review of "Rapid saturation of cloud water adjustments to shipping emissions" by Manshausen et al. (egusphere-2023-813)**

The authors have addressed my comments carefully, and I fully support the manuscript's publication in Atmospheric Chemistry and Physics. Below, I state a few suggestions that the authors may want to consider.

**Suggestions** (all line numbers refer to the tracked changes document)

A recent paper by Diamond (2023) also addresses the impact of shipping fuel regulations on clouds. I suggest referencing this publication.

Ll. 7 – 9: Instead of writing about "increases", I suggest writing about "anomalies". I associate a "constant increase" with a constant dLWP/dNd or dLWP/dNa, which are not independent of emissions, as they would result in an increase of LWP proportional to Nd or Na.

L. 21: Replace "they" with "these", or remove "studies".

L. 43: Define "IR" or replace with "infrared".

L. 54: Change to "The mixing of the track with the surrounding air, […]".

Ll. 118 – 119, Fig. 6a: Any ideas why the Nd anomaly increases with free-tropospheric dryness? Could this be due to an enhanced suppression of precipitation by the evaporation of LWP via the entrainment of drier air?

L. 123: "controlling", not "controllong".

**References**

Diamond, M. S. (2023). Detection of large-scale cloud microphysical changes within a major shipping corridor after implementation of the International Maritime Organization 2020 fuel sulfur regulations. *Atmospheric Chemistry and Physics*, *23*(14), 8259-8269.